# Behavioral contagion on social media: Effects of social norms, design interventions, and critical media literacy on self-disclosure

**Philipp K. Masur**[1]*, **Dominic DiFranzo**[2], **Natalie N. Bazarova**[3]

**1** Department of Communication Science, Vrije Universiteit Amsterdam, Amsterdam, Netherlands, **2** Department of Computer Science and Engineering, Lehigh University, Bethlehem, Pennsylvania, United States of America, **3** Department of Communication, Cornell University, Ithaca, New York, United States of America

\* p.k.masur@vu.nl

**Data Availability Statement:** The data underlying the results presented in the paper are available from the following Open Science Framework Project Page (https://osf.io/qxjsp/).

## Abstract

Social norms are powerful determinants of human behaviors in offline and online social worlds. While previous research established a correlational link between norm perceptions and self-reported disclosure on social network sites (SNS), questions remain about downstream effects of prevalent behaviors on perceived norms and actual disclosure on SNS. We conducted two preregistered studies using a realistic social media simulation. We further analyzed buffering effects of critical media literacy and privacy nudging. The results demonstrate a disclosure behavior contagion, whereby a critical mass of posts with visual disclosures shifted norm perceptions, which, in turn, affected perceivers' own visual disclosure behavior. Critical media literacy was negatively related and moderated the effect of norms on visual disclosure behavioral intentions. Neither critical media literacy nor privacy nudge affected actual disclosure behaviors, however. These results provide insights into how behaviors may spread on SNS through triggering changes in perceived social norms and subsequent disclosure behaviors.

## 1 Introduction

People are susceptible to social influence through behaviors of others around them [1–3]. The behavioral influence can extend both in positive and negative directions: either nudging ethical and responsible actions, personal growth, and deliberation, or, in contrast, discouraging rationality, forming bad habits, and increasing risk factors [4]. Norm effects whereby people copy common behaviors occur even when people are aware that norms are arbitrary and may not reflect other people's actual preferences [5].

In online environments and in particular social network sites (SNS), it has become widespread to share private information publicly with other users and often by extension, with third parties, service providers, and institutions [6]. While some disclosure of personal information is inevitable because of online platforms' terms of use and in order to reap benefits of SNSs [7], high levels of online self-disclosure can be problematic, both from an individual and

**Funding:** This project was funded by NSF CHS grant ##1405634 awarded to Natalie Bazarova. Recruitment was funded within the Collaborative Project "Digital Behaviors in the Digital Age" of the Cornell Center for Social Science. The funders had no role in study design, data collection and analysis, decision to publish, or preparation of the manuscript.

**Competing interests:** The authors have declared that no competing interests exist.

a societal perspective [8]. Sharing too much information online may invite a risk of horizontal (i.e., stemming from other users) and vertical privacy violations (i.e., stemming from online service providers or institutions [9]), information exploitation and commodification [10], ubiquitous monitoring and surveillance [11], and privacy devaluation overall [12].

A normative pressure for online disclosure stemming from prevalent behaviors on SNSs is especially problematic in a rapidly evolving media landscape, which requires privacy vigilance and critical engagement. Limited knowledge about data collection practices, how shared information is used and by whom, and risks involved with information flows in interconnected media environments, contribute to a feeling of uncertainty about online privacy and disclosure [6]. In such a state of uncertainty, it is natural to look to other people's behaviors for guidance about appropriate disclosure, especially in situations where the risk of social sanctioning is high, as in publicly observable behaviors [13]. Furthermore, if prevalent sharing of private information online indeed creates a social norm for online disclosure, there is even a greater cause for concern for how it can be employed by online service providers vying for users' attention, time, and sharing on their platforms. An algorithmic curation of search results and SNSs' feeds that prioritizes posts with high levels of self-disclosures can propagate more disclosures through online social networks, similar to the emotional contagion effect [14].

The goal of this paper is to understand the role of prevalent behaviors containing visual disclosures of self-identifying pictures (the collective norm) in shaping perceived social norms and, in turn, other users' behaviors on SNSs. A corollary goal is to explore potential buffering factors—critical media literacy and design nudges—that may counteract a normative influence for more self-disclosure and instead promote more privacy-aware behavior. If successful, these buffers could be leveraged to foster deliberate and self-reflective posting behaviors through media literacy education and platform design alterations. To this end, we conducted two pre-registered studies that provide a comprehensive picture of the relationships between the amount of others' visual disclosure (0%, 20%, 80% in Study 1 and 5% and 80% in Study 2), perceived norms, and participants' own visual disclosure behavioral intentions (Study 1) and observed behaviors (Study 2). By employing a realistic social media simulation, Study 2 allowed for the observation of behavioral adaption over a 2-day period to the experimentally manipulated disclosure behavior prevalence in a naturalistic online environment.

## 2 Study 1

### 2.1 Defining social norms

Social norms are "rules or standards that are understood by members of a group, and that guide and/or constrain social behavior without the force of law" [4]. Whereas *collective* norms refer to an established code of conduct (i.e., the actual prevalence of a behavior) and operate at the societal level or the level of the social network, *perceived* norms capture individual perceptions of behaviors and pressures to conform [1, 2]. Perceived norms can be further differentiated into descriptive, injunctive, and subjective norms. *Descriptive* norms refer to a perceived prevalence of a behavior, which can deviate considerably from its actual prevalence captured by a collective norm (e.g., as in the case with a pluralistic ignorance [15]). They often serve as a shortcut that people rely on, especially in new and unfamiliar situations, when they lack information or guidance on what constitutes an appropriate behavior [4]. *Injunctive* norms refer to individuals' belief about which behavior others approve of [4]. This belief reflects a moral judgement about what *ought* to be done [1].

Yet, there are situations, in which descriptive and injunctive norms do not overlap. For example, although many people think that organ donation is a noble deed (injunctive norm), far fewer of them volunteer to be organ donors through state organ-donor registries

(descriptive norm) [16]. A third type of norm is a *subjective* norm, which refers to the perceived pressure to enact a certain behavior [2, 17]. In contrast to perceptions of what is done and what ought to be done, it reflects perceptions of of what others *expect* a person to do [16]. A subjective norm exerts a strong influence on behaviors because of because of people's desire to avoid risking interpersonal harmony by going against others' expectations [2].

## 2.2 The influence of social norms on self-disclosure

Privacy research has explored norm perceptions as one of many antecedents of self-disclosure and privacy behaviors, often drawing from the theory of planned behavior [17]. Survey studies suggest a small- to medium-sized relationship between social norms and privacy-related outcomes, with larger effects obtained when there was a close alignment between the measured norms and behaviors [18–21]. For example, there has been found a strong correspondence ($\beta$ = .40) between a privacy protection subjective norm (i.e., what others expect one to do for privacy protection) and an intention to engage in privacy protection behaviors [22]. Furthermore, peer influence was a significant predictor of privacy behavior in the network analysis of Facebook users in the US: If participants' network of Facebook friends generally had private profiles, they were more likely to have private profiles as well [23]. In line with communication privacy management theory [24], injunctive norms of peer groups can also inhibit self-disclosure, according to a qualitative study of German Facebook users [25].

Despite these promising results, previous research has a few limitations that render the norm-self-disclosure link inconclusive. Most studies have examined the relationship between perceived norms and self-reported behavior, with limited attention to the effects of prevalent behaviors (i.e., the collective norm) on self-disclosure. Furthermore, no study, to the best of our knowledge, examined norm influences on actual observed behaviors, rather than self-reports. This latter limitation could be especially consequential because of methodological issues connected to the use of self-reported behaviors. First, highly compatible self-report measures of norms, intentions, and behaviors are likely to correlate due to their similarity. Second, self-reported communication behaviors suffer from recall and self-report inaccuracies, as evidenced by recent comparisons between self-reports and logged media [26]. These two problems combined cast doubts on how the prevalence of a certain behavior translates to actual behavior and how strong norm effects actually are on behaviors.

In this study, we explicitly focused on *visual disclosure* on SNSs as sharing selfies has become an essential part of the social media experiences [27]. We define visual disclosure as sharing a photo of a person's own face and thereby making oneself identifiable. While this is a commonplace practice, a perpetuating social norm cycle of sharing pictures of oneself and adapting to this norm can increase both overt (e.g., identity theft, cyberstalking, misuse, recontextualization) and hidden privacy risks and violations (e.g., unwittingly sharing meta data such as when and where an image was taken;) [28]]. Our first goal was thus to investigate the causal effects of the collective norm related to visual disclosure—operationalized as the proportion of posts with a photo of a person's face—on norm perceptions and behavioral intentions. Given the strong evidence of norm effects in prior research [19–22], we hypothesized:

> H1: A higher proportion of posts with users showing themselves in the feed (the collective norm) leads to a higher a) descriptive, b) injunctive, c) subjective norm and d) a stronger intention to share a picture of oneself on the site.

That said, there may be other factors in a person's communicative environment that can amplify or attenuate the effect of social norms related to visual disclosure. In relation to visual

disclosures, people encounter visual disclosures in other users' profiles, in addition to their posts. As a defining feature of almost any SNS platform [29], profile pictures are linked to almost any interaction or communicative act. SNS profiles can be regarded as "good examples of online settings where intimate story telling is practiced" [30]. Prior research has already suggested the "profile work," which includes strategic self-presentation via profile pictures, may be influenced by social norms as well as influence social norm building [31]. Yet, due to a lack of prior research on how identifiable profile pictures affect sharing norms, we pose the following research questions:

> RQ1: Do the a) descriptive, b) injunctive, c) subjective norm and d) the intention to share a picture of oneself increase when more users show themselves in their profile pictures?

> RQ2: Does the proportion of profiles with users' identifiable photos of themselves moderate the effects of the proportion of posts with users showing themselves in the feed on the a) descriptive, b) injunctive, c) subjective norm and d) the intention to share a picture of oneself?

## 2.3 Preventing mindless adaption to risky social norms

As outlined above, sharing information without considering privacy risks and consequences can be deeply problematic. Therefore, another goal of this study was to understand counteracting factors that could serve as an antidote against social normative influences of potentially risky disclosure behaviors. Online privacy literacy and general media literacy, in particular, may be such a factor that could enhance independent thinking and self-determination in information exchange in online environments [32, 33]. This is supported by the positive association between greater awareness of online service providers' data collection and the active use of online data protection strategies [34, 35].

Media literacy research generally distinguishes between knowledge (structures), self-reflection abilities and personal locus, and actual skills related to accessing, analyzing, evaluating, and creating media content across a variety of contexts [32, 36]. An important aspect of media literacy refers to critical media literacy skills, which may be defined as an ability to criticize, question, and challenge existing assumptions about media content, the ways in which media content is produced, and the motives and goals behind the creation of media content [37, 38]. Individuals with high critical media literacy are more motivated and competent participants of social life and know "how to use media as instruments of social communication and change" [38]. For example, higher critical media literacy skills can prevent a negative effect of idealized body representations in social media [38, 39]. Critical media literacy skills may play a similar buffering role in attenuating the effect of collective norm on visual disclosure behaviors. In a similar vein, a general disposition towards critical thinking, "reasonable reflective thinking focusing on deciding what to believe or do" [40] could mitigate the effects of the collective norm by increasing the propensity for self-correction of intuitions and behaviors [41]. Given the lack of research on buffering factors in relation to self-disclosure, we posed the following research question:

> RQ3: Does a) critical social media literacy and b) critical thinking disposition reduce the effects of the collective norm on the intention to share a picture of oneself?

## 2.4 Methods

Our goal was to move beyond investigating the norm-behavior link using self-reported survey data. Findings from cross-sectional surveys only allow to identify between-person relationships of self-reports of perceived norms and retrospective, aggregated behavioral measures, also based on self-reports. Although differences in norm perceptions have been systematically linked to differences in disclosure behavior in such studies, it remains open whether other users' self-disclosure behaviors on the site (i.e., the collective norm) causally *lead* to different norm perceptions, and in turn, *affect* subsequent self-disclosure intentions and actual self-disclosure behavior.

To investigate such causal effects, we aimed to develop a research design that allows for a combination of realism and control using a SNS simulation implemented within a fully functional SNS called EatSnap.Love (ESL; for previous work based on this research design, see https://socialmedialab.cornell.edu/the-truman-platform/). This platform looks similar to Instagram, but invites users to share food-related posts (see Fig 1). Users are able to sign up and create an account, scroll through a feed of food-related posts, share pictures of themselves, and like or respond to post of other users. Within this platform, an additional implementation called *Truman* (named after the 1998 film *The Truman Show*) allows a simulation of a realistic and operational, but fully controlled social media environment, which is identical for all participants by experimental conditions. In this simulated environment, participants observe, interact, and communicate with other users that they believe are real human beings, but which are actually *bots* programmed to behave and respond according to the experimental conditions. In a first step, Study 1 presented participants with a snapshot of the simulation, which contained about 50 posts and respective comments, and investigated people's subsequent disclosure intentions. In a second step, Study 2 (see further below) implemented the fully operational simulation over a two-day period and investigated people's actual behavioral adaptation.

Both studies were run using Amazon's Mechanical Turk (MTurk). MTurkers are demographically more diverse than the traditional university participant pool or standard Internet samples [42] and perform better on attention checks and experimental instructions [43]. Recent research has further shown that the data quality is similar to data obtained from graduate or commercial participant pools [44].

Data, materials, analysis syntaxes of both studies as well as reproducible versions of this manuscript and the online supplement material (OSM) can be accessed via https://osf.io/qxjsp/. The OSM includes all item formulations, descriptive analyses, reliability and factor analyses, and additional analyses: https://osf.io/rqft5/. The software to run ESL and Truman is open source. It can be downloaded and used for scientific purposes via https://github.com/difrad/social_norms_truman.

**2.4.1 Procedure, power, and sample.** The hypotheses, design, and analysis plan for Study 1 was preregistered on the 5th of February 2019 (https://osf.io/ufmnv/; small deviations—e.g., rewording of the hypotheses for consistency—from the preregistrations of Study 1 and 2 are logged here: https://osf.io/dytaz/). We based our minimum sample size on power calculations for the main effects of the experimental manipulations and a potential interaction between both. Assuming a smallest effect size of interest of f = .15 for main effects and interaction effects on both norm perceptions and the likelihood of disclosing oneself resulted in a minimum sample size of $N$ = 536 (given a significance level of 5% and a power of 80%).

Data was collected between the 6th and 7th of February 2019. Overall, $N$ = 677 people were eligible for the study. We excluded 29 participants who did not spend at least 10 seconds looking at the social media feeds. We further excluded 15 participants because they didn't pass a simple attention check. After listwise deletion of missing values, the final sample size was

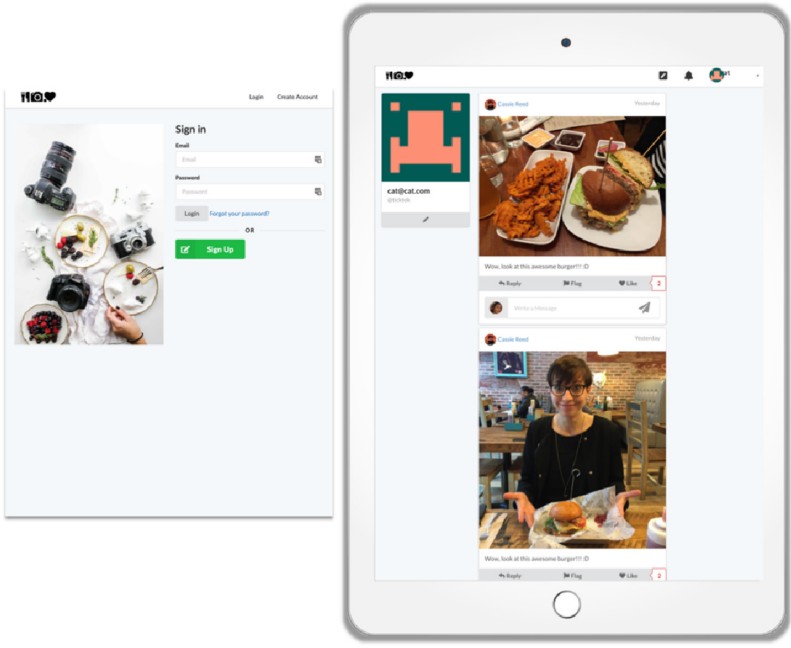

**Fig 1. The social network site EatSnap.Love (left: Sign in page; right: Example of how the platform and the feed looks like to the user).** The amount of posts in which the users disclosed themselves (an example is the lower post in the feed) was manipulated. In study 1, either 0%, 20% or 80% of the posts show the face of the user, and either 0%, 20% or 80% of the profile pictures showed the user. In Study 2, either 5% or 80% of the posts showed the user.

$n$ = 576 ($M$ = 36.9 years ($SD$ = 11.6, range = 19–87; 59.4% female; 4% bachelor or college degree). Completion of the survey took on average $M$ = 12.0 ($SD$ = 8.2) minutes, with a compensation of $0.50 for the study completion.

**2.4.2 Experimental manipulations.** Participants were asked to scroll through a snapshot of the SNS simulation (~50 posts and respective comments, see Fig 1). The cover story was that we wanted to test a new social network site and explore its usability and overall feel. We manipulated the collective norm by varying the amount of visual disclosure in posts and the profile pictures. The study thereby followed a 3 (0%, 20%, or 80% of the posts revealed the face of the user) x 3 (0%, 20%, or 80% of the users had an identifiable profile picture) between-subject design. Participant were randomly assigned to one of nine groups ($n$ = 58–70 per group). Groups did not differ in sample size, age, gender, or education. The randomization was hence successful. To ensure external validity, we tested whether the conditions were perceived as realistic. We found that participants viewed the condition with 80% of the profile pictures and 0% of the posts containing photos of a person's face as slightly less realistic compared to participants who viewed the condition with 80% of the profile pictures and 80% of the posts with photos of a person's face, $p$ = .007. Overall, however, all conditions were perceived as highly realistic (marginal means were between 5.33 and 5.95; scale ranging from 1 to 7), rendering this small difference negligible. After participants were confronted with the respective simulated feed, we assessed the following measures.

**2.4.3 Measures.** To measure participants' *social norm perceptions* we adapted the 12-item scale developed by Park and Smith [16] to fit the simulated SNS environment ESL. Four items referred to descriptive (e.g., "Most people on EatSnap.Love are visually identifiable in their posts"), injunctive (e.g., "The majority of people on EatSnap.Love think it is appropriate to share pictures of themselves."), and subjective norm perceptions (e.g., "I have the feeling that

most people on EatSnap.Love want other users to post pictures of themselves") respectively and were administered on a 7-point scale ranging from 1 (*strongly disagree*) to 7 (*strongly agree*). The three-dimensional model ($\chi^2(51) = 343.74$, $p < .001$; CFI = .97; TLI = .96; RMSEA = .10, 90% CI [.09, .11]; SRMR = .02) fitted the data well, but all three subdimensions correlated very strongly ($r > .90$), suggesting that these all three factors did not have enough discriminant validity. A second-order model revealed that all three types of norms loaded highly onto a global factor ($\gamma > .90$). We hence continued our analyses with a single factor (and a single mean index respectively). Reliabilities of all three subdimensions were high ($\omega = .93$-.95). The reliability of the whole scale was likewise very high ($\omega = .98$).

The *likelihood of disclosing oneself* in posts on the platform was measured with a self-developed scale with six items (e.g., "If I would use EatSnap.Love, I would share pictures in which I can be identified"), on a 7-point scale ranging from 1 (*strongly disagree*) to 7 (*strongly agree*). The uni-dimensional model fitted the data well ($\chi^2(5) = 15.89$, $p = .007$; CFI = >.99; TLI = .99; RMSEA = .06, 90% CI [.03, .10]; SRMR = .01) and the reliability was high ($\omega = .97$).

We developed a new item pool (20 items) for *critical media literacy* by adapting items from the more comprehensive media literacy scales by Tamplin et al. [45], Zhu et al. [46], and Koc and Barut [47] (e.g., "I think about what social media companies do to get my attention.") as well as by adapting items from the deliberation subscale by Costa and McCrae [48] (e.g., "I rarely make hasty decisions on social media"). All items were measured on a 7-point scale ranging from 1 (*strongly disagree*) to 7 (*strongly agree*). Factor analyses revealed three subdimensions (see pp. 10-14 in the OSM). The first dimension captured the ability to "reflect consequences before posting on social media." The second dimension consisted of the adapted deliberation items and thus captured "being deliberate when using social media." The last dimension reflected a "critical information literacy" in general terms. The three-dimensional model fitted the data well ($\chi^2(42) = 131.02$, $p < .001$; CFI = .96; TLI = .94; RMSEA = .06, 90% CI [.05, .07]; SRMR = .05) and the subdimensions correlated moderately with critical thinking disposition ($r$ between .23 and .56), suggesting good convergent validity. The reliabilities of the subscales were good ($\omega = .77$-.81).

*Critical thinking disposition* was measured with eight items that were based on Baron et al.'s cognitive reflection test [49], Yoon's critical thinking disposition scale [50], and Cacioppo et al.'s need for cognition scale [51]. Participants answered the questions on a 7-point scale ranging from 1 (*strongly disagree*) to 7 (*strongly agree*). After removing two items, the model fitted the data well ($\chi^2(9) = 38.21$, $p < .001$; CFI = .96; TLI = .94; RMSEA = .08, 90% CI [.05, .10]; SRMR = .04) and reliability of the scale was good ($\omega = .77$).

**2.4.4 Data analysis.** Due to a low amount of missing values (0.2%), we deleted cases with missing values listwise. In line with our preregistration, we created mean indices for all variables in line with the CFA models described above. We tested the individual main effects and interactions using separate analyses of variance (ANOVAs). We used Tukey's honestly significant difference (HSD) to tests subsequent pairwise comparisons. All moderation analysis were computed with continuous moderators, but findings were additionally exemplified using a pick-a-point approach (see Fig. 3).

## 2.5 Results

In line with our predictions, the data revealed a strong effect of the collective norm (i.e., having posts that contained users' faces in the activity feed) on social norm perceptions ($F(2,567) = 342.81$, $p < .001$, $f = 1.100$; see Fig 2). Based on Tukey's honestly significant difference (HSD) tests, there was a significant difference across all levels of posts with faces presented in the activity feed (0%, 20%, and 80%). Interestingly, however, the difference in social norm

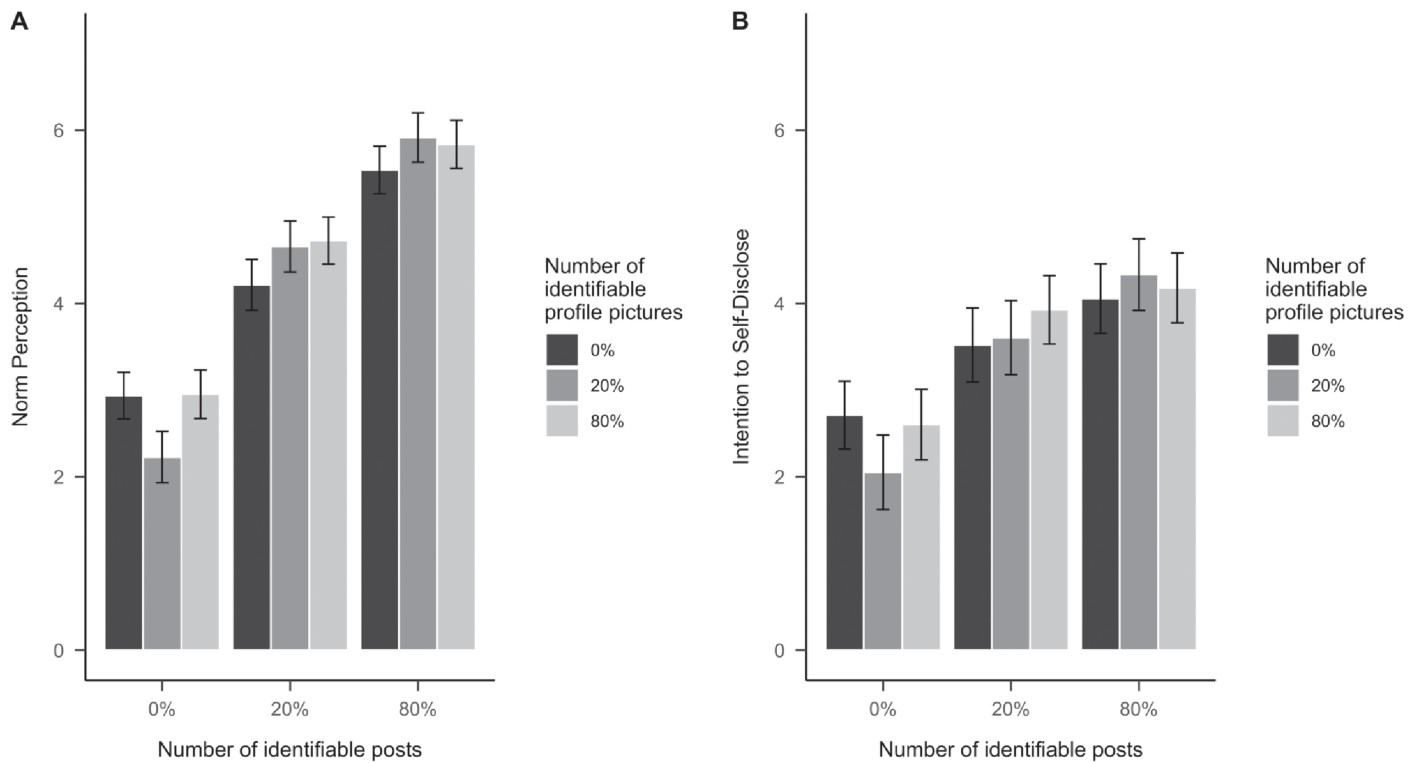

**Fig 2.** The effect of number of identifiable profile pictures and number of identifiable posts on norm perceptions (A) and intentions to self-disclose (B) in Study 1. Bars represent estimated marginal means. The error bars represent 95% confidence intervals.

perceptions was larger between 0% and 20%, compared to 20% vs. 80% (Fig 2A). H1a-c were hence supported.

The percentage of posts with faces in the activity feed likewise had a moderate, positive effect on the intention to disclose oneself on the platform ($F(2,567) = 53.97$, $p <.001$, $f = 0.436$; see Fig 2B). H1d was hence supported. However, only the differences between 0% and 20% and between 0% and 80% were significantly different, suggesting a small occurrence of events, even a minority—20% of all posts in this case, may provide enough of a tipping point to trigger changes in behavioral intentions. Based on the joint significance approach [52], it can further be concluded that norm perceptions mediate the effect of the collective norm on disclosure intention. Specifically, the effect of the collective norm on intention to disclose oneself became non-significant ($F(2,554) = 1.74$, $p = .176$, $f = 0.079$), when the norm perceptions were entered into the model first, which positively predicted the intention to self-disclose ($F(1,554) = 297.75$, $p <.001$, $f = 0.733$).

With regard to RQ1, we found that the amount of identifiable profile pictures only had a very small, yet significant effect on norm perceptions ($F(2,567) = 3.04$, $p = .049$, $f = 0.104$; see Fig 2A) and no significant effect on the intention to disclose oneself ($F(2,567) = 0.94$, $p = .393$, $f = 0.057$; see Fig 2B).

There was also a small, but significant interaction between both factors on norm perception ($F(4,567) = 5.19$, $p <.001$, $f = 0.191$), in response to RQ2. That said, due to the small effect sizes, the direct and moderating effects of the amount of visual disclosure in profile pictures were deemed negligible.

With regard to RQ3, we investigated whether any of the three subdimensions of critical media literacy (deliberate posting, reflection ability, and critical information literacy)

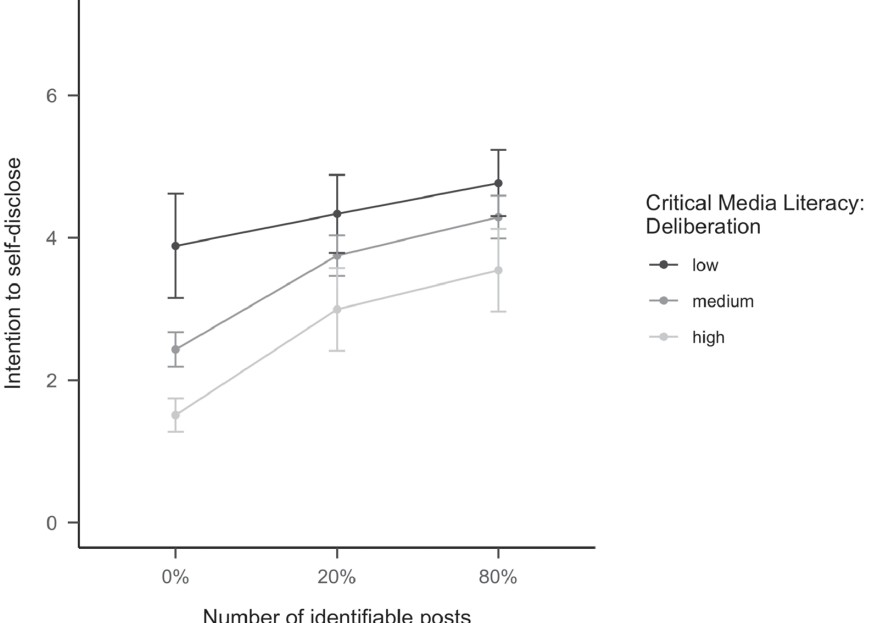

**Fig 3. Critical media literacy (deliberate social media use) influences the effect of number of identifiable posts on norm perceptions on the intention to self-disclose (Study 1).** The error bars represent 95% confidence intervals. The moderation analysis was conducted using the continuous moderator. Here, we exemplify the moderation effect by showing exemplary results for participants with medium (= average critical media literacy in the sample), high (= M + SD), and low (= M—SD) levels of literacy.

moderated the effect of the collective norm on self-disclosure intentions. Out of the three dimensions of critical media literacy, only deliberate posting was negatively associated with visual disclosure behavioral intention ($F(1,555) = 65.53$, $p <.001$, $f = 0.344$). Furthermore, deliberate posting also moderated the effect of the collective norm distribution on the intention to visual disclosure ($F(2,555) = 5.07$, $p = .007$, $f = 0.135$). Contrary to our assumptions, however, it did not reduce the effect of the collective norms on intention to self-disclose. Instead, individuals with low levels on this dimension indicated that they would be quite comfortable in sharing photos of themselves, *independent* of the collective norm condition they were randomly assigned to ($Ms = 3.89$–$4.77$; no significant differences based on HSD tests, see also Fig 3).

In contrast, individuals with medium or higher levels of that critical media literacy were responsive to the visual norm distribution (see Fig 3) showing the lowest intention to visually disclose themselves when no other users did so in their posts (0%: $M_{medium} = 2.43$; $M_{high} = 1.51$), but they adapted to the norm when 20% (20%: $M_{medium} = 3.75$; $M_{high} = 2.99$) or more posts (80%: $M_{medium} = 4.29$; $M_{high} = 3.55$) showed faces in the feed. Yet even then, they still reported a lower intention to self-disclose compared to participants with a low critical media literacy. General critical thinking disposition neither directly affected the intention to self-disclose nor moderated the effect of the collective norm on disclosure intentions.

## 2.6 Discussion of Study 1

Summarizing the results of Study 1, we found that visual norm distribution of posts with users' faces led to different social norm perceptions, and, in turn, affected intentions to disclose themselves visually. It is noteworthy that the effect of the collective norm manipulation became evident even with a minority of norm-bearing posts in the feed (20%), and no difference in

behavioral intention emerged between 20% and 80% norm manipulations. The amount of profile pictures with users' faces, in contrast, had a negligible effect on norm perceptions and disclosure intentions. Our results further suggest that overall critical media literacy does not mitigate the effect of the collective norm on self-disclosure, but people with specific critical media literacy skills such as deliberation abilities are more aware of their social environment and appear to adapt to it more deftly compared to those with lower levels who reported a high intention to disclose themselves regardless of the visual disclosure norm. While these results are promising, they only captured the effects of collective norms on behavioral intentions, rather than behaviors, which suggests a need for further replication and an in-depth analysis based on actual behavior.

## 3 Study 2

### 3.1 Replicating findings with behavioral measures

With Study 2, we wanted to test whether the results from Study 1 could be replicated in a simulated SNS environment, in which we could observe actual behavioral adaptations to the norm prominence in situ. Building on the causal effect of the collective norm of visual disclosure on both norm perceptions and visual disclosure intentions found in Study 1, we hypothesized:

> H1: A higher proportion of posts with users showing themselves (the collective norm) leads to a higher a) descriptive, b) injunctive, c) subjective norm and d) and more sharing of posts with one's own picture on a SNS.

Study 1 further suggested that deliberate posting skills as a subdimension of critical media literacy worked as a buffer against indiscriminate sharing of visual disclosure, but instead, participants who scored high on these skills appeared to be more deliberate in their adaptations to social norms. Building on this finding, we hence hypothesized:

> H2: Deliberate posting is a) negatively related to sharing of posts with one's own pictures, but b) is positively related to adapting one's visual disclosure to the collective norm.

### 3.2 Buffering normative influences through nudging

Although media literacy has always been regarded as a fundamental skill in self-determination and deliberation, a "do-it-yourself" data protection approach puts the main burden on an individual user [53]. It is questionable whether an average user can actually accomplish the task of understanding and recognizing all risks and challenges related to privacy in an increasingly complex and ever-changing media environment. A viable alternative may be offered through design interventions, such as subtle privacy nudges [54] that can prompt users to engage in more deliberate sharing [55–57]. A well-designed privacy nudge may help individuals overcome decision-making biases and reliance on heuristics (e.g., loss aversion, optimism bias, anchoring, bounded rationality) by presenting more contextual cues, prompting reflection, and allowing for reversibility of mistakes [55]. Thus, H3 predicts the effects of a privacy nudge on resistance to collective norms:

> H3: A nudge asking users to reflect on their decision to post a picture on the site reduces a) sharing of posts with one's own picture and b) the normative effect of seeing posts with pictures of other users on one's visual disclosure behavior.

To further substantiate the effect of the collective norm on behavior, we controlled for variables that prior research has shown to be related to self-disclosure in general and visual disclosure in particular. Privacy calculus theory argues that online self-disclosure depends on rational cost-benefit calculations [58], and privacy concerns and perceived benefits of social media use have been found to be oppositional predictors of self-disclosure on social media [59, 60]. We hence controlled for, on the one hand, horizontal and vertical privacy concerns [9] and, on the other hand, perceived gratifications of social media use such as relationships initiation and maintenance, enjoyment, and self-presentation [59].

### 3.3 Methods

**3.3.1 Procedure, power, and sample.** Study 2 was preregistered on the 5th of April 2019 (https://osf.io/xb4pv/; small deviations from the preregistration of Study 2 are logged here: https://osf.io/dytaz/). The sample size was based on power calculations for the significance level of 5%, a power of 80%, an effect size of $f = .20$ for the main and interaction effects (half of the found effect size in Study 2: $f = .40/2 = .20$) resulting in a targeted sample size of $n = 277$ (assuming a continuous outcome variable). Given that the outcome variable (coded behavior) is binary, we aimed to recruit a larger sample size (15% more: $n > 396$).

Data was collected between the 24th of April 2019 and the 17th of May 2019, with the same cover story of testing a new social network site to explore its usability and overall feel. Overall, out of 1,202 MTurkers who clicked on the study, 553 participants completed all parts of the study (pre-survey, using the site for at least one day, post-survey). We excluded another 7 participants because they did not provide consent to use all their data and another 7 because they failed to pass two simple attention checks. The final sample consisted of $n = 526$ ($M = 35.20$ years, $SD = 10.70$, range = 18-71; 58.17% female; 31.94% college or bachelor degree). Participants were compensated $10 for completion of all study parts ($0.50 for completion of pre- and post survey each, $3 per day and another $3 for full completion).

**3.3.2 Experimental manipulations.** We manipulated both the prevailing collective norm related to visual disclosure in posts and the design of the SNS to include a privacy nudge. The design therefore followed a 2 (5% vs. 80% of the posts revealed the face of the user, see Fig 1) x 2 (reflection nudge vs. no nudge, see Fig 4) between-subject design. As Study 1 showed that no faces in post was perceived the least realistic (although still realistic), we opted for ratios that would both be perceived as similarly realistic. The nudge was designed to both allow users to reverse their decision (i.e., to recover from suboptimal decision), as well as to present them with additional contextual cues [i.e., the potential audience, see 54]. Participants were hence randomly assigned to one of four groups ($n = 123-145$). Again, groups did not differ in sample

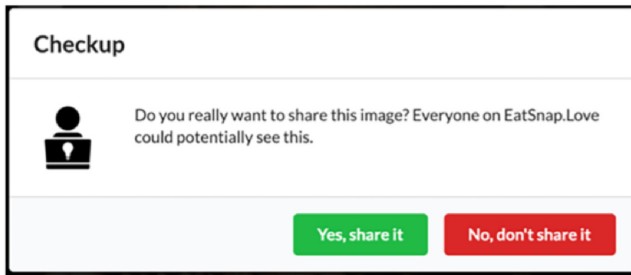

**Fig 4. The nudge implemented in EatSnap.Love.** In the privacy nudge condition, participants were confronted with this additional checkup window after they clicked 'submit' to post a photo. In the no privacy nudge condition, the post was directly published after participants clicked 'submit'.

size, age, gender, or education. The randomization was hence successful. All four conditions were again perceived as realistic (norm manipulation: $F(1,520) = 0.77$, $p = .380$; nudge manipulation: $F(1,520) = 2.31$, $p = .129$; $Ms = 5.41$–$5.60$; scale ranging from 1 to 7).

During the two-day "testing phase" of ESL, each participant saw other "users" which were in fact automated "bots" programmed to create new posts at certain intervals, and to comment on the posts created by other "users." Participants were led to believe that they were using the site with many other active human users. No participant was able to see the activity or behaviors of any other participant, or interact in any way with another participant. Thus, the participants will see only interactions between other "bots."

Once participants consented to participate in the study, they were redirected to an onboarding process that instructed them how to use ESL and a pre-survey with socio-demographic and media use-related questions. During the 2-day study period, they were instructed to publish at least one post each day and to interact with other users (who were actually bots) through replying, liking, and commenting on posts presented in an "activity feed," for at least two 5-minute periods each day. At the end of the two-day period, participants were asked to complete a post-study survey, were debriefed with an online debriefing form, compensated for their participation, and were given the option of having their data removed from analysis.On average, participants visited the site M = 7.88 (SD = 6.71) times over the course of the study.

**3.3.3 Person-related measures.** We used the same *social norm perception scale* as in Study 1 as we were uncertain with regard to whether the convergence of all three norms in one global factor would replicate when participants were able to actually use the platform and experience the feed over a period of two days. However, despite the multidimensional scale fitting the data well, the subdimensions again correlated strongly (*r* between.58 and.85). To avoid multi-collinearity issues or suppression effects, we again decided to use the second-order factor in our subsequent analysis (i.e., all items formed one global norm perception). The second-order factor fitted the data well ($\chi^2(51) = 162.09$, $p < .001$; CFI = .98; TLI = .97; RMSEA = .06, 90% CI [.05, .08]; SRMR = .04). The reliabilities of the subscales were good ($\omega = .87$-.95).

To assess *critical media literacy*, we used the three-dimensional scale developed in Study 1. The three-dimensional model again showed good factorial validity ($\chi^2(51) = 176.01$, $p < .001$; CFI = .95; TLI = .93; RMSEA = .07, 90% CI [.06, .08]; SRMR = .06) and all subscales showed good reliabilities ($\omega = .77$-.81).

The *online privacy concerns* scale [9] included five dimensions of which two can be subsumed under vertical (i.e., concerns with regard to online service providers and institutions) and and three under horizontal privacy concerns (i.e., concerns with regard to unwanted access of other users, unwanted information sharing by other users, identify theft by other users). An example items is e.g., "How concerned are you about social media companies sharing the information you posted on social media with third parties?" Participants answered 3 items per subdimension on a 7-point scale ranging from 1 (*not at all concerned*) to 7 (*very concerned*). In the CFA, the second-order model fitted the data well ($\chi^2(84) = 366.50$, $p < .001$; CFI = .97; TLI = .96; RMSEA = .08, 90% CI [.07, .09]; SRMR = .06). All subdimensions had high reliablities (.91-.97).

The perceived *benefits of social media use* scale [59] distinguishes motives related to relationship maintance, enjoyment, and self-presentation with three items per subdimensions (e.g., "Social media allow me to make a better impression on others"), all measured on a 7-point scale ranging from 1 (*strongly disagree*) to 7 (*strongly agree*). The three-dimensional model fitted the data well ($\chi^2(24) = 132.14$, $p < .001$; CFI = .97; TLI = .95; RMSEA = .09, 90% CI [.08, .11]; SRMR = .04) and all dimensions showed high reliabilities ($\omega = .82$-.94).

**3.3.4 Data analysis.** Although the amount of missing data was very low (0.1%), a listwise deletion would have removed participants who provided valuable data on the within-person

level from the two-day testing phase (e.g., if they chose to not report their age, they would have been deleted despite providing valuable log data). Missing values were hence imputed using a multiple imputation approach (5 fully imputed data sets). However, we only report results from one of the imputed data sets as they did not differ from the pooled results (for pooled results across all 5 imputed data sets, see p. 38 in the OSM). We computed mean indices for all variables in line with the CFA results described above. All independent variables were centered around the grand mean.

To obtain a measure for the dependent variable of visual disclosure, we coded the pictures participants posted during their 2-day participation on the site. The first author (blind with regard to which condition participants were in) coded each picture into identifiable (1 = *showed the discloser*) or non-identifiable (0 = *did not show the discloser*) for all of participants' posts. Posts containing pictures of participants' children or younger siblings were also coded as "identifiable" (a deviation from our preregistration, in which we planned to only code pictures of the discloser as "identifiable"), as disclosing the identity of a relative or a child can also identify the discloser.

To test H1a-H1c, we tested the main between-person effects of the experimental manipulations on norm perception using linear regression models and their effects on the overall self-disclosure behavior (per person sum indices based on the coded posts), as well as their interaction effect on overall self-disclosure behavior using negative binomial regression (due to the coded count data). However, of those who did disclose themselves in posts, the majority only disclosed themselves (or their children or relatives) once. We hence decided to estimate a logistic regression model due to the low variance in the coded data (for a comparison of results, see p. 40 in the OSM). Originally, we also wanted to explain whether participants disclosed themselves on the post-level (cf. preregistration). However, due to the low within-person variance, a two-level logistic multilevel model in which visual disclosure measurements (the coded behaviors) are nested in persons did not converge (see OSM).

## 3.4 Results

In line with H1a to H1c and the findings from Study 1, participants perceived a stronger social norm in the condition in which 80% of the post disclosed other users' faces compared to when only 5% disclosed themselves ($b = 1.58$, $se = 0.08$, 95% CI [1.43, 1.74], $p < .001$, $\beta = .66$, see Fig 5), explaining 43% in the variance of the social norm perceptions.

Next, Table 1 presents results from logistic regression models predicting the observed disclosure behavior (the coded behavior). In line with H1d and replicating the finding of Study 1, the collective norm had a positive and significant effect on disclosure behavior (Model 3: $b = 1.69$, $se = 0.58$, Odds Ratio = 5.40, 95% CI [1.75, 16.71], $p = .003$). In other words, individuals who were in the condition in which 80% of the posts revealed the user were 5.40 times more likely to share a picture that showed themselves. This effect became non-significant (Model 4: $b = 0.92$, $se = 0.65$, Odds Ratio = 2.51, 95% CI [0.70, 9.02], $p = .157$), when norm perceptions were entered into the model, which positively predicted disclosure behavior (Model 4: $b = 0.51$, $se = 0.23$, Odds Ratio = 1.66, 95% CI [1.06, 2.61], $p = .028$). Based on the joint significant approach, this suggests that the effect of the collective norm on disclosure behavior is partially mediated by norm perceptions.

The implemented privacy nudge had a negative, but non-significant effect on disclosure behavior (Model 3: $b = -0.16$, $se = 0.78$, Odds Ratio = 0.85, 95% CI [0.18, 3.95], $p = .839$). H2a was hence not supported. In contrast to H2b, it further did not lower the effect of the collective norm manipulation on disclosure behavior either (Model 3: $b = 0.14$, $se = 0.87$, Odds Ratio = 1.15, 95% CI [0.21, 6.30], $p = .876$). Deliberate social media use (or any other subdimension of

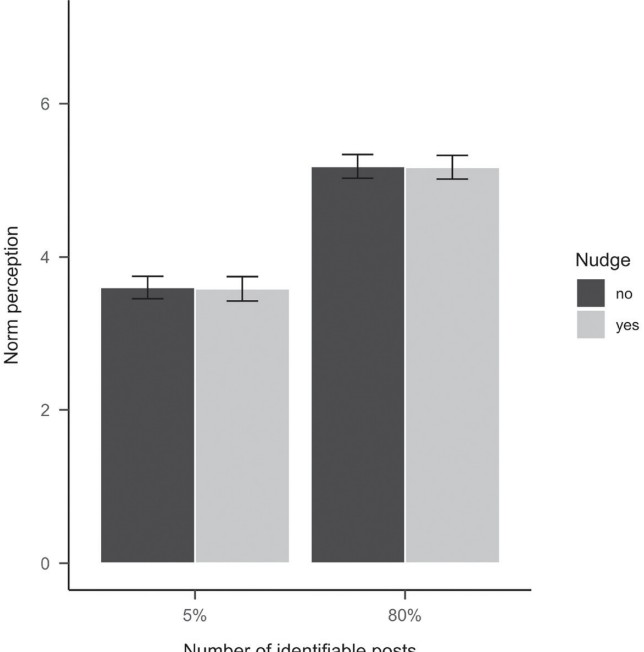

**Fig 5. The effect of number of identifiable profile pictures and number of identifiable posts on norm perceptions in Study 2.** Bars represent estimated marginal means. The error bars represent 95% confidence intervals.

critical media literacy) neither negatively predicted disclosure behavior (see Table 1), nor moderated the effect of collective norms on disclosure ($b = -0.10$, $se = 0.37$, Odds Ratio = 0.90, 95% CI [0.44, 1.85], $p = .784$; for these analyses, see Table 43 in the OSM), in contrast to H3 and H4 and the findings from Study 1.

## 3.5 Discussion of Study 2

Summarizing the results from Study 2, we found that in line with Study 1, the amount of posts with users' faces led to different social norm perceptions. The more users disclosed themselves visually, the higher was the perceived social norm to disclose oneself visually. This, in turn, led to more actual visual disclosures (albeit only 8.33% of the participants actually made themselves visually identifiable). In contrast to Study 1, critical media literacy neither affected visual self-disclosure nor moderated the effect of the collective norm on behavior. Finally, our results did not support the privacy nudging effect: Prompting SNS users to reflect on their decision to share a photo did not reduce a rate of visual disclosure and did not mitigate the norm effect either, possibly because of the low occurrence of visual disclosures, as noted above.

## 4 Overall discussion

The goal of this research was to provide an in-depth analysis of how social norms affect self-disclosure on SNSs. We argued that norm effects on self-disclosure might be particularly strong in today's online environments because the affordances of SNSs make other users' behaviors more perceptible and thus the deduction of norms easier [61]. Building on social norm theory [2], we designed two consecutive studies that provide distinct, yet complementary insights. This research extends prior work by confronting participants with different collective norms and measuring their subsequent norm perceptions and intentions to disclose (Study 1) and their behavioral adaptations in response to the manipulated norms in situ over a

**Table 1. Logistic regression model predicting visual disclosure (Study 2).**

| | Model 1 | Model 2 | Model 3 | Model 4 |
|---|---|---|---|---|
| (Intercept) | -3.05 (0.32) *** | -3.10 (0.33) *** | -4.20 (0.60) *** | -4.02 (0.61) *** |
| Control Variables | | | | |
| Age (in Years) | -0.00 (0.02) | -0.01 (0.02) | -0.01 (0.02) | -0.01 (0.02) |
| Gender (1 = female) | 0.81 (0.36) * | 0.80 (0.37) * | 0.82 (0.38) * | 0.86 (0.39) * |
| Education | 0.03 (0.14) | 0.04 (0.14) | 0.04 (0.15) | 0.07 (0.15) |
| Privacy Calculus Variables | | | | |
| Vertical Privacy Concerns | | -0.23 (0.14) | -0.26 (0.15) | -0.24 (0.15) |
| Horizontal Privacy Concerns | | 0.09 (0.16) | 0.09 (0.17) | 0.08 (0.18) |
| Motiv 1: Relationships Maintenance | | -0.08 (0.25) | -0.11 (0.26) | -0.13 (0.27) |
| Motiv 2: Enjoyment | | 0.23 (0.22) | 0.26 (0.22) | 0.22 (0.23) |
| Motiv 3: Self-Presentation | | -0.15 (0.23) | -0.15 (0.24) | -0.18 (0.25) |
| Experimental Manipulations | | | | |
| Collective Norm (# of Identifiable Posts) | | | 1.69 (0.58) ** | 0.92 (0.65) |
| Privacy Nudge (0 = No; 1 = Yes) | | | -0.16 (0.78) | -0.00 (0.79) |
| Interaction Between Manipulations | | | 0.14 (0.87) | 0.09 (0.88) |
| Critical Media Literacy | | | | |
| Responsible behavior | | | | 0.13 (0.23) |
| Deliberate Media Use | | | | -0.13 (0.15) |
| Information Literacy | | | | -0.37 (0.25) |
| Social Norm Perception | | | | |
| Descriptive, Injunctive, and Subjective | | | | 0.51 (0.23) * |
| Pseudo $R^2$ (McFadden) | 0.02 | 0.04 | 0.11 | 0.14 |
| Pseudo $R^2$ (Nagelkerke) | 0.03 | 0.05 | 0.14 | 0.18 |
| Pseudo $R^2$ (Cox & Snell) | 0.01 | 0.02 | 0.06 | 0.07 |
| AIC | 285.38 | 290.89 | 275.32 | 274.91 |
| BIC | 302.44 | 329.27 | 326.50 | 343.16 |
| Log Likelihood | -138.69 | -136.44 | -125.66 | -121.46 |
| Deviance | 277.38 | 272.89 | 251.32 | 242.91 |

*Note.* Unstandardized coefficients (standard errors) are reported. $n$ = 526.

*** $p < 0.001$;

** $p < 0.01$;

* $p < 0.05$

2-day period (Study 2). Because social norms for sharing information could expose users to privacy risks and violations, we further investigated two potential buffering factors: critical media literacy as a person-related characteristic and a privacy nudge as an easy-to-implement design intervention. The studies followed open science best practices through the pre-registration of hypotheses and analysis plans as well as making the data, material, and analysis scripts publicly available [62, 63].

## 4.1 Conceptual and methodological contributions

Although conceptually different (as evidenced by well-fitting multi-dimensional models), descriptive, injunctive, and subjective norms related to self-disclosure emerged as congruent across both studies. This may suggest that in novel environments (as represented by EatSnap. Love), people heavily rely on descriptive norms to infer injunctive and subjective norms. SNS affordances [61] may increase the perceptibility of the collective norm. The persistent nature

of SNS posts provides users with a broad basis for inferring descriptive norms. Likes and comments on posts offer an immediate feedback about audiences' level of approval, which is useful for making inferences about injunctive and subjective norms. At the same time, SNS affordances have expanded opportunities for self-disclosure and self-disclosure gratifications yet also complicating the management of risks and unwanted consequences: while posts might easily reach larger audiences online and thus provide an ideal platform for self-presentation and self-expression [7], the broad accessibility of one's disclosure heightens the risk of privacy violations and recontextualization [9]. Subsequent studies should thus systematically investigate the ways in which feedback and social cues affect subsequent norm perceptions.

Future research should further look into the role of time and experience on the platform for developing norm perceptions, for example, differences between newcomers vs. old-timers in how they perceive what others do (descriptive) vs. what they approve of (injunctive) vs. what they expect them to do (subjective). Understanding critical points for different norms' convergence may open up opportunities for interventions to nudge users' behaviors based on how they view and internalize behaviors, moral judgments, and subjective expectations around them.

Other important findings concern the role of social norms in explaining people's disclosure behaviors. Even after controlling for variables that are key to the privacy calculus theory, social norms still explained a substantial part of the variance in self-disclosure (Study 2). This highlights the importance of incorporating social norms into privacy and self-disclosure theories, as social norm perceptions may serve as heuristic shortcuts, as opposed to rational cost-benefit calculations based on the privacy calculus. The integration of rational and automatic processes would yield a more comprehensive picture of how people self-disclose on social media. A related finding from Study 1 suggests that collective norms (others' actual behaviors) do not need to be prevalent to facilitate a normative change: a minority of posts (around 20%) was sufficient to instill perceived social norms about behaviors and their appropriateness. This result aligns with the recent finding about a critical mass of 25% of the population as a tipping point to initiate a change in social conventions and social dynamics [64].

With regard to potential factors that might reduce the effects of social norms and thereby foster more privacy-aware behavior, our studies yield mixed findings. Study 1 suggests that deliberate social media skills can negatively predict self-disclosure intentions and potentially moderate the effect of the collective norm on self-disclosure intentions. That said, we were not able to replicate this finding in Study 2, when we observed actual behavior. On the one hand, this underlines the importance of differentiating intentions from actual behaviors. On the other hand, if a moderation effect of critical media literacy indeed exists, it is most likely to be very small. In fact, it is probably too small to be detected if a dependent variable has a low variance (in Study 2, only very few people disclosed themselves and even fewer participants disclosed themselves more than once). The low variance in self-disclosure in Study 2 could also be the reason why the implemented privacy nudge did not work. Although it offered participants an opportunity to reverse their decision to disclose a photo and gave them a reminder about a potential viewing audience, the variance in self-disclosure might have simply been too small to detect an already small effect of nudges on behavior. Future studies should investigate whether differences in the presentation of nudges (e.g., size, layout, colors) or the severity of the presented consequences (e.g., real-life consequences) might increase the impact of nudges on people's privacy awareness and behavioral response, especially for common disclosure behaviors.

Furthermore, in line with research on the discrepancy between self-reports and tracking behavior [26] we believe that present findings highlight the importance of studying actual behavior. Privacy studies and online communication research, in general, suffers from a strong

methodological focus on cross-sectional survey studies relying on self-reports. EatSnap.Love and Truman offer a research platform for creating an ecologically valid social media experience that is nonetheless closely controlled by the researcher. By sharing the platform with the research community, it is our hope that researchers will use Truman to move beyond the limitations of traditional methods. The possibility to manipulate simulated online worlds (e.g., varying the type or amount of posts displayed to participants, changing the design of the site, etc.) allows for a realistic, yet controlled investigation of a variety of different online phenomena. As mentioned before, one direction could be to investigate how a typical interaction (likes, comments, and shares) affects perceptions of norms in social media. Although not manipulated in this study, it seems likely that users of SNSs infer the injunctive norm not only by observing what others do, but also by evaluating which posts receive positive feedback and social validation in the form of likes, shares, and positive comments. We thus strongly advocate for developing and using innovative methods that allow for the unbiased observation on online communication.

From a methodological perspective, we finally urge future research to test the found effects with more diverse samples. Although MTurk samples are not necessarily of lesser quality than samples from other commercial panels, concerns remain with regard to biases resulting from the experiences of professional Mturkers with deception and scientific paradigms [65].

## 5 Conclusion

This is one of the first research studies to examine a behavioral contagion effect experimentally by observing how others' behaviors beget different perceived norms, which, in turn, shape an observer's own disclosure behaviors. By running a consecutive set of studies that build on each other, we have explored normative influences on behavioral intentions and behavioral adaptations in real time in response to others' behaviors on the platform. The results demonstrate a strong link between others' behaviors, perceived norms, and one's own behaviors, even in networks with strangers, and that it only takes a minority of the population to shift perceived norms and behaviors online.

## Author Contributions

**Conceptualization:** Philipp K. Masur, Dominic DiFranzo, Natalie N. Bazarova.

**Data curation:** Philipp K. Masur, Dominic DiFranzo.

**Formal analysis:** Philipp K. Masur.

**Funding acquisition:** Natalie N. Bazarova.

**Investigation:** Philipp K. Masur, Dominic DiFranzo, Natalie N. Bazarova.

**Methodology:** Philipp K. Masur, Dominic DiFranzo, Natalie N. Bazarova.

**Project administration:** Philipp K. Masur, Dominic DiFranzo.

**Resources:** Natalie N. Bazarova.

**Software:** Philipp K. Masur, Dominic DiFranzo, Natalie N. Bazarova.

**Supervision:** Natalie N. Bazarova.

**Validation:** Philipp K. Masur, Dominic DiFranzo, Natalie N. Bazarova.

**Visualization:** Philipp K. Masur.

**Writing – original draft:** Philipp K. Masur.

**Writing – review & editing:** Philipp K. Masur, Dominic DiFranzo, Natalie N. Bazarova.

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
