## [Decision Letter · Decision Letter 0]

9 Jun 2021

PONE-D-21-13829

Behavioral Contagion on Social Media: Effects of Social Norms, Design Interventions, and Critical Media Literacy on Self-Disclosure

PLOS ONE

Dear Dr. Philipp K. Masur,

Thank you for submitting your manuscript to PLOS ONE. Our reviewers just shared their evaluation and acknowledge that your submission is one of great interest and quality. Congratulations! That said, you are asked to submit a revised version with the requested clarifications (see comments below: most of them are related to the method and the discussion of the results).

We look forward to receiving your revised manuscript.

Kind regards,

Jonathan Smith, Ph.D.

Academic Editor

PLOS ONE

Journal Requirements:

3) Please ensure that you refer to Figure 4 in your text as, if accepted, production will need this reference to link the reader to the figure.

4) Please include a copy of Table 2 which you refer to in your text on line 512.

Reviewers' comments:

Reviewer's Responses to Questions

**Comments to the Author**

1. Is the manuscript technically sound, and do the data support the conclusions?

Reviewer #1: Yes

Reviewer #2: Partly

2. Has the statistical analysis been performed appropriately and rigorously? 

Reviewer #1: Yes

Reviewer #2: Yes

3. Have the authors made all data underlying the findings in their manuscript fully available?

Reviewer #1: Yes

Reviewer #2: Yes

4. Is the manuscript presented in an intelligible fashion and written in standard English?

Reviewer #1: Yes

Reviewer #2: Yes

5. Review Comments to the Author

Reviewer #1: The communication and influence of social norms on computer-mediated environment is an interesting topic for scholars but is also a gap in the current literature. This manuscript used data from two experiments to address this gap with a focus on how perceived norms of disclosure on SNS affects self-disclosure behavior. The experiments were rigorously conducted. The presentation of theory and data was lucid. Overall, it is a solid piece of work that has potential to advance the online social influence literature. I have some minor observations as below, which I hope would help to improve the manuscript:

1. Page 2: "....In such a state of uncertainty, it is natural to look to other people's behaviors for guidance about appropriate disclosure." Is this conditioned by whether or not communicators' identities are anonymous? Perhaps, anonimity might reduce the influence of uncertainty on the link between norms - behavior. People might not be too concerned about the opinions and attitudes of others in the anonymity condition because their own identities are hidden and therefore, their need for social proof is low while social sanction is also unlikely. Several studies related to online incivility might provide some evidence to this possible effect. So, perhaps the authors should consider clarifying this argument a bit?

2. Page 2: Individual-norm types: Are subjective norms similar to injunctive norms? In the reasoned action approach, they seem to be used interchangeably. As the data showed these norms did not discriminate, if the authors operationalize norms as two dimensions of descriptive norms and injunctive norms, what would the CFA results be like? In Study 2, there was a correlation r = .58, which was moderately strong and thus, norm types might still be discriminate?

3. Page 2-3: Definition of collective norms: Does the concept necessarily connect to a reference group? Rimal and Lapinski (2015) posit that collective norms operate at the societal level and that social identity works as a moderating factor in the norm-behavior link. The reference group likely involves social identification. I wonder if the reference group as operationalized in this study truly captures the population-level collective norms of users using the social media site as tested, unless the reference group is manipulated to saliently represent the large collective users. To be more specific, would the exposure to 50 posts as presented in this study capture the actual code of conduct of a large population of users? If not, the causal effect of collective norms on normative perception as currently claimed might not be clear. Perhaps, mentioning that participants observing other users' posts and comments would suffice as you did on page 4, line 163.

4. Page 3, line 109: "there may be other factors..."

5. Page 6, line 213, remove bracket.

6. What were the differences between posts and comments? How were the comments created and what did they say? Were there possible effects of these comments in the experiment?

7. Measures: Can you add a few sample items for other norm types?

8. Can you please explain why the 2 items measuring critical thinking were removed? Is it because of low factor loading (e.g., below .40)?

9. In Study 1, it looks like you emphasized the literature method gap of not measuring actual behavior, but Study 1 did not account for this gap. You only did so in Study 2. So, would it be more appropriate to move the argument to the front end and explain in Study 1 that you aimed at examining norms-likelihood of self-disclosure (or behavioral intention as you wrote on page 8, line 328) as the first step of the project?

Other than that, I feel that the manuscript is clear and a pleasure to read. The recommendation for future research is very informative. I'd also recommend emphasizing the specific social interactive network platforms because research has shown that heuristic features such as like, share, and comments might work differently depending on different platforms and user groups. Thank you for the opportunity to read this qualified work.

Reviewer #2: The authors present a duo of consecutive and complementary studies investigating the effects of social norms on intended and actual disclosure behaviors on a fictional social media site. The first study tests intentions to visually disclose on the site based on reviewing approximately 50 posts. They find a relatively small percentage (20%) of posts are enough to influence perceptions of social norms encouraging visual disclosures. Additionally, the critical media literacy subconstruct of Deliberation moderates the effect of social norms on intentions to visually disclose. The second study builds on this by inviting participants to spend two days using the platform and captures their activity – primarily visual disclosures. While the findings with social norms is replicated, Deliberation was not a significant factor in Study 2.

Most of my research focuses on online self-disclosure, so I am quite familiar with this area of study. I’ve also worked with MTurk for data collection. I currently have more experience with survey methods than experiments, but I am familiar with experiments.

Overall, I enjoyed reviewing this research study. I found the research questions and the approach to the study interesting. The paper is quite polished in terms of writing, with only a handful of minor errors. The discussion is reasonable and doesn’t overextend the findings of the two studies. I do think there is room for improvement, though many of my recommendations are to help make parts of the manuscript clearer and to add some additional detail in other places. The points below are organized by my perception of most pressing to address.

1. Why was Study 2 only over two days? What evidence indicates this would be sufficient time to elicit an adequate number of disclosures? It seems very odd for it to be so short, and thus unsurprising that most participants only had a single visual disclosure.

2. Why was ANOVA the chosen statistical method for Study 1? Given the nature of the data, SEM or regression (as in Study 2) seems more appropriate. It isn’t clear why ANOVA was selected or what benefit it provides over other statistical approaches.

3. Study 1 uses nine different groups and Study 2 uses four groups based on the combination of manipulations. The authors don’t address in the manuscript if these different groups are statistically similar to each other. As such, it isn’t clear if there are other factors that can be attributed for the differences observed.

4. In both studies while discussing the collective norm effect, I find myself confused. After some reflection, I’ve determined it’s because I was thinking of the collective norm and the social norm perceptions as being the same constructs. I don’t readily associate the manipulations in the study with the term collective norm. Even in studying Table 1, I wasn’t associating “Number of Identifiable Posts” with collective norm or the manipulation. I know the authors spend some time identifying the different social norms on Page 2, but collective norms receive comparatively little discussion. It gets challenging to keep up with the different norms, especially when they’re all social norms and that term is most often only used in reference to perceived norms. I think consistency would help prevent this confusion, among other clarifying strategies.

5. Why was the same perceived norm scale used in Study 2? Alternatively, why were the three perceived norms hypothesized separately in Study 2? Study 1 provided evidence that the selected measure lacked discriminant validity for the three subscales, so it seems illogical to do the same thing and expect different results. I believe this needs to be addressed directly either with the hypothesis or the measure discussion. That said, I appreciate the discussion at the end of the paper and proposed future research regarding this issue.

6. The manuscript presents three research questions on page 4 – well into setting up and discussing the first study. I generally prefer research questions to appear earlier in the paper (i.e. the introduction) because I view these as big picture questions that inform the model (and hypotheses by extension) and research design. It’s confusing to me that these are presented after a hypothesis, particularly as the text leading up to the RQs seems to indicate a specific hypothesis – before the authors seem to lose confidence and indicate there’s insufficient extant research to support a specific hypothesis. I think presenting the research questions earlier (especially as they relate to both studies) and then discussing these other avenues of exploration as ways to address the research questions would be more direct and allow the research questions to serve as overall guides for the manuscript; this also allows the authors to speculate about the possible relationship based on what is known without forming specific hypotheses.

7. It isn’t clear if the likelihood of disclosing oneself scale is self-developed. If it is, why were existing scales not used/modified and what pretesting was done to validate the measure?

8. Relatedly, it isn’t clear if the 20-item pool of critical media literacy items was self-developed or drawn from the referenced studies. Additionally, it isn’t clear if the deliberation subscale was added to the 20-item pool or counted within it. The language in this part just needs some clarifying.

9. Section 4.1, Lines 581-583, the authors discuss visual disclosure in Study 1 as though it represents behavior and is therefore directly comparable to the actual behaviors captured in Study 2. However, my understanding is that Study 1 asked respondents how likely they would be to post information on the simulated site after reviewing several posts; I interpret this as intention to act in a certain way, not actual behavior. As such, it may be that Deliberate Media Use is a significant predictor of intention but not behavior, which is why it’s significant in Study 1 but not Study 2. I agree that low variance may also be a factor in detecting small effects, but I also think it’s important to consider what exactly has been captured in each study regarding visual disclosure.

10. Beginning in Study 2, the authors reference SNS. To this point – with the exception of the abstract – the authors have referred to social media. These are not interchangeable concepts (SNS is the more narrowly defined of the two). Considering Instagram and Twitter are the referenced sites for the simulated environment, it may be inaccurate to call it an SNS – Twitter is a microblog, not an SNS, and it seems like the experimental setting may mimic this more because friend lists (a key component of SNS) are not mentioned or seem likely to be supported.

11. To that last point, it isn’t clear why relationship management was included as a perceived benefit. Relationship management indicates that there’s a pre-existing relationship to nurture. This simulated environment isolates users so they only interact with bots so – by extension – they don’t know the other “users” and therefore can’t have pre-existing relationships to manage. It seems illogical to include.

12. The authors state that posts containing children and other family members were coded as identifiable and then explains why this kind of photo deidentifies the target individual. First, this discussion makes me question if there’s a typo – e.g. that the photo wasn’t coded identified because more people in the picture deidentifies the target person – so I think that should be revisited for clarity. Second, it’s noted that this differed from the pre-registration of the study but it isn’t explained why.

13. I think a table summarizing the hypotheses and results by study (i.e. hypotheses on the rows and studies on the columns) would be useful going into the final discussion.

14. In Study 1, participants were excluded for spending less than 10 seconds on the simulated feed; how did the authors determine how long participants spent on the simulated feed?

15. Similarly, in Study 2, it’s stated that participants were asked to spend at least 5 minutes on the platform each day. How was this tracked? What was the average time spent on the site each day?

16. The authors do a fair job justifying use of an MTurk sample. However, there are several studies that raise valid concerns with collecting data on MTurk – including the problem of professional MTurkers. I didn’t see any formal limitations of the study discussed, but it might be worth at least mentioning using a different sample in future research to avoid potential biases caused by only using MTurk samples.

17. Section 3.3.1, Lines 391-393, the authors mention using the “same cover story,” but no cover story was discussed for Study 1.

18. The authors make a statement about the shortcomings of cross-sectional surveys that is uncited (Section 2.4, Lines 159-161).

19. Figure 1A is quite blurry, so it’s difficult to see what exactly is going on. It might be worthwhile making A and B separate figures (especially as 1B is only relevant to Study 2) so that the images of 1A can be slightly larger and hopefully clearer.

20. Very minor editing. The most consistent error is not spelling out numbers zero through nine. There are also a couple of times within the Study 2 section where a comparison is being made to Study 1 but the authors have said Study 2 instead (e.g. Lines 433 and 438).

~*~

Separately – although I don’t think much can be done about it now – I wanted to raise my concern with participant compensation in this duo of studies. In both studies, the participants seem grossly underpaid for their participation. MTurk typically recommends paying participants the equivalent of minimum wage in the United States ($7.25 per hour). While the authors don’t specify in the manuscript if a location filter was used, the compensation on the first study is $2.50 an hour for the average participant. While payment looks better for Study 2 ($10 total), the average time to complete the pre- and post-surveys was not discussed, so it’s harder to judge if compensation was reasonable. There isn’t much that can be done on this point now, but I wanted to raise this issue because I find it unethical to so poorly compensate participants.

6. PLOS authors have the option to publish the peer review history of their article (what does this mean?). If published, this will include your full peer review and any attached files.

Reviewer #1: No

Reviewer #2: No

---

## [Author Response · Author response to Decision Letter 0]

28 Jun 2021

Dear Dr. Smith,

Dear Reviewers,

We would like to thank you for your thorough and thoughtful comments. We carefully addressed all of them. Please see our detailed responses to each of the reviewers’ comments in the attached file "Response to Reviewers". We highlighted the parts with more profound changes in yellow in the manuscript (Revised Manuscript with Track Changes). 

Please also note that the Open Science Framework (OSF) page related to this manuscript (https://osf.io/qxjsp/) was updated slightly as well (note that the current manuscript refers to study 2 (https://osf.io/yqhjr/) and 3 (https://osf.io/sxp2f/) within the broader project and the reproducible version of the manuscript as well as the respective online supplement is located in the folder “Manuscript 2 & Online Supplement 2” (https://osf.io/32zxe/). These pages contain a lot of additional information, including all data, scripts, and materials to reproduce/replicate the results/study as well as additional analyses. For some answers below, we refer to the relevant documents respectively. 

Once again, thank you for your thoughtful comments. It is our hope that we addressed all of your comments satisfactorily.

Sincerely,

The Authors

---

## [Editor Report · Decision Letter 1]

1 Jul 2021

Behavioral Contagion on Social Media: Effects of Social Norms, Design Interventions, and Critical Media Literacy on Self-Disclosure

PONE-D-21-13829R1

Dear Dr. Philipp K. Masur,

We’re pleased to inform you that your manuscript has been judged scientifically suitable for publication and will be formally accepted for publication once it meets all technical requirements.

Kind regards,

Jonathan Smith, Ph.D.

Academic Editor

PLOS ONE

---

## [Editor Report · Acceptance letter]

5 Jul 2021

PONE-D-21-13829R1 

Behavioral Contagion on Social Media: Effects of Social Norms, Design Interventions, and Critical Media Literacy on Self-Disclosure 

Dear Dr. Masur:

I'm pleased to inform you that your manuscript has been deemed suitable for publication in PLOS ONE. Congratulations! Your manuscript is now with our production department. 

Kind regards, 

on behalf of

Dr. Jonathan Smith 

Academic Editor

PLOS ONE